# Artificial Intelligence in the Advanced Diagnosis of Bladder Cancer-Comprehensive Literature Review and Future Advancement

**DOI:** 10.3390/diagnostics13132308

**Published:** 2023-07-07

**Authors:** Matteo Ferro, Ugo Giovanni Falagario, Biagio Barone, Martina Maggi, Felice Crocetto, Gian Maria Busetto, Francesco del Giudice, Daniela Terracciano, Giuseppe Lucarelli, Francesco Lasorsa, Michele Catellani, Antonio Brescia, Francesco Alessandro Mistretta, Stefano Luzzago, Mattia Luca Piccinelli, Mihai Dorin Vartolomei, Barbara Alicja Jereczek-Fossa, Gennaro Musi, Emanuele Montanari, Ottavio de Cobelli, Octavian Sabin Tataru

**Affiliations:** 1Department of Urology, IEO—European Institute of Oncology, IRCCS—Istituto di Ricovero e Cura a Carattere Scientifico, 20141 Milan, Italy; 2Department of Urology and Organ Transplantation, University of Foggia, 71121 Foggia, Italy; 3Urology Unit, Department of Surgical Sciences, AORN Sant’Anna e San Sebastiano, 81100 Caserta, Italy; 4Department of Maternal Infant and Urologic Sciences, Policlinico Umberto I Hospital, Sapienza University of Rome, 00161 Rome, Italy; 5Department of Neurosciences and Reproductive Sciences and Odontostomatology, University of Naples Federico II, 80131 Naples, Italy; 6Department of Translational Medical Sciences, University of Naples “Federico II”, 80131 Naples, Italy; 7Urology, Andrology and Kidney Transplantation Unit, Department of Emergency and Organ Transplantation, University of Bari, 70124 Bari, Italy; 8Department of Urology, ASST Papa Giovanni XXIII, 24127 Bergamo, Italy; 9Department of Oncology and Hemato-Oncology, University of Milan, 20122 Milan, Italy; 10Department of Urology, Medical University of Vienna, 1090 Vienna, Austria; 11Division of Radiation Oncology, IEO—European Institute of Oncology IRCCS, 20141 Milan, Italy; 12Department of Urology, Foundation IRCCS Ca’ Granda—Ospedale Maggiore Policlinico, 20122 Milan, Italy; 13Department of Clinical Sciences and Community Health, University of Milan, 20122 Milan, Italy; 14Department of Simulation Applied in Medicine, George Emil Palade University of Medicine, Pharmacy, Science and Technology of Târgu Mures, 540142 Târgu Mures, Romania

**Keywords:** artificial intelligence, machine learning, deep learning, diagnosis, bladder cancer

## Abstract

Artificial intelligence is highly regarded as the most promising future technology that will have a great impact on healthcare across all specialties. Its subsets, machine learning, deep learning, and artificial neural networks, are able to automatically learn from massive amounts of data and can improve the prediction algorithms to enhance their performance. This area is still under development, but the latest evidence shows great potential in the diagnosis, prognosis, and treatment of urological diseases, including bladder cancer, which are currently using old prediction tools and historical nomograms. This review focuses on highly significant and comprehensive literature evidence of artificial intelligence in the management of bladder cancer and investigates the near introduction in clinical practice.

## 1. Introduction

Artificial intelligence (AI) has been an explored area since the rise of computers and software and studies as the human brain and its abilities to learn from experience, quickly adapt to a novel setting, imagine and work with abstract concepts, and manipulate the surrounding environment [1]. Nowadays, we are all speaking about AI, not just in the scientific and research settings, mesmerizing us all with the ability of AI to interact with our day-to-day activities such as social networks, smart devices, driving cars, and chat interactions with AI software [2,3]. When it comes to medical sciences, cancer research, specifically bladder cancer (BCa), is benefiting from AI applications for the current advanced possibility of diagnosis of this neoplasia. AI applications for the diagnosis of BCa integrate imaging with bladder segmentation, tumor detection on cystoscopy, tumor staging, and tumor grading [4]. Both AI subsets represented by machine learning (ML) and deep learning (DL) techniques are heavily studied in regard to diagnosis, prognosis, and prediction of outcomes of BCa [4,5,6]. In this review of the literature, we aimed to comprehensively analyze the existing literature that focuses on the advancements of BCa diagnosis. First, we described in an easily understandable fashion the concept of AI and specific terms used focusing on ML and DL. Secondly, we reviewed the range of AI applications in BCa diagnosis possibility. Lastly, we discuss the perspective of AI in the diagnosis of BCa and the challenges that still need improvements for a wide clinical integration in optimal cancer care.

## 2. Materials and Methods

We have developed a comprehensive review focusing our research on the improvement in the knowledge of AI, to familiarize with AI applications, to establish further potential research in the advanced diagnosis of BCa, and to explore the possibility of entering in clinical practice of these applications. For this purpose, we have searched the PubMed database to identify original articles on the mentioned topics from the last ten years. Keywords used were “artificial intelligence”, “machine learning”, “diagnosis”, “deep learning”, and “bladder cancer”. We have included articles up to March 2023 exploring any methods or modalities of bladder cancer diagnosis, regardless if it was the first or the follow-up diagnosis (recurrence or progression). The available research has been screened by article title and abstract by two independent reviewers, and the included evidence has been further interpreted after the approval of all authors.

## 3. Results

By searching the databases, we have found 94 research papers. No duplicate files were removed, and 94 abstracts were screened for eligibility. Forty-eight met the criteria for analysis using “artificial intelligence”, “machine learning”, “diagnosis”, “deep learning”, and “bladder cancer” as keywords (Figure 1). After a full-text examination, all 48 papers were ultimately included.

### 3.1. Generalities of Artificial Intelligence

Since the beginning, several definitions for AI have been established, but the main element in these descriptions are the capabilities of a machine or computer’s ability to mimic the human brain and cognition with the purpose of elaborating the optimal strategy for the desired outcome [7,8,9].

#### 3.1.1. Machine Learning

ML is an AI subset and computer science that uses enormous amounts of data and, through mathematical and statistical algorithms aims to replicate the functionality of the human brain. ML has the ability to build classification models or to make predictions based on the data it is trained on [10]. The way it uses data and its algorithms is definitory as ML does not receive pre-defined inputs from the surrounding environment in order to learn and make predictions [7,8] and is trained to understand data (e.g., images, numbers) and the connection between input variables. The process is based on examples, gained experience, and if given instructions, ML can learn autonomously [11].

ML holds the potential to analyze outcomes without being explicitly pre-programmed and uses data to learn how to perform a given task. This sub-branch of AI can generate algorithms that can analyze data to predict outcomes. The interesting ability to, automatically and without further human intervention, adapt its own programming in order to reach the given task is a basic feature of ML [12]. Computer-aided systems use the ML algorithm methods (from input features to output variables) to discover correct variable values from new, previously undiscovered features. The automated learning process uses vast amounts of data that has to be of sufficient quality to render good results. The methodology basically incorporates training, validation, and test datasets. We have this flow of the process pictured in (Figure 2).

The problems of classification and regression of ML are due to the supervised learning category, which means that the computer-aided diagnosis (CAD) system is fed with datasets controlled by a human specialist. The annotations of the human specialists are mandatory and pre-designated to the system as benign or malignant (e.g., histopathology, imaging (computed tomography (CT), magnetic resonance imaging (MRI), cystoscopy images) of BCa lesions. Therefore, the interaction between machine learning experts and medical researchers is desired to be an efficient process for compiling the necessary features [4,12].

#### 3.1.2. Deep Learning

Deep learning is a type of ML that relies on deep artificial neural networks. A basic artificial neural network (ANN) has interconnected nodes that connect with nodes from the next layer using edges. These edges give the interconnecting strength and can be measured. A DL network has an input, an output, and in between several hidden layers [13]. As a result, it does not need engineers to develop o model for learning [14]. The interconnecting nodes are called perceptrons, which are a super-simplified version of a biological neuron that takes different inputs and weighs them up to produce a single output [15]. Using specific algorithms, hidden layers fine-tune the data to minimize error, and the activation function gives the output data, and the ANN produces its result [16]. DL has the potential to fully automate tasks based on the neural architecture of the human brain using multi-layer neural network algorithms, and this enables DL to solve computational problems such as image classification [15], which holds the promise to add value to the BCa diagnosis and detection [17,18]. A type of ANN used to digitize images is a convolutional neural network (CNN) because it recognizes patterns [19]. CNNs are great for identifying and classifying images and visual recognition of problems in dermatology [20], ophthalmology [21], and oncology [22]. Segmentation of images from imaging platforms such as urography CT, MRI, and cone beam CT using deep learning models has been published lately as it can be used to stage the primary tumor.

A deep neural network is designed to incorporate a vast amount of interconnecting computed and computing neurons that have been constructed layer after layer [23,24]. Every neuron receives data from the neurons in the layer before and sends data to neurons in the consecutive layer. All consecutive layers are also called hidden layers and receive training data with the ground truth. This data goes into a process of multiplication, divination, addition, and subtraction and then is transmitted to the output layer and offers the prediction. Supervised learning is often used in medicine and uses images. CNNs used in oncology have archived great performance, similar to human experts [25]. The imaging data (lines, curves, different colors) are fed into the first layers of the algorithm; afterward, the higher-level layers are retrained to offer diagnostic predictions [26].

Numerous DL approaches combine genomic, transcriptomic, and histopathological data. The aim is to enhance patient diagnosis, prognosis, and treatment. The human expert place still remains essential in oncology (clinicians have the ability to analyze data in the clinical context), and DL is complementary to disease research [5,27]. The potential of utilizing a large number of variables and parameters makes DL a good strategy for predicting outcomes. A limitation is related to overfitting (i.e., adaptation to the background noise that is unique to each sample). The generalization for new patient populations is therefore limited and is typically healthcare where economics, data security, and patient privacy issues often limit data availability [28]. Particularly in BCa, different DL methods have been used in BCa diagnosis settings (bladder segmentation, tumor detection, tumor staging, and grading), broadly by using bladder cystoscopy, different imaging techniques, histopathology, and cancer genomics [4,12]. A depiction of how DL is working can be found in Figure 3. The summarization of the terms generally used to shortly describe deep learning is embedded in Table 1.

### 3.2. Bladder Cancer Diagnosis

#### 3.2.1. Bladder Tumor Detection through Cystoscopy

Cystoscopy still plays an important role in tumor detection (in primary diagnosis and follow-up settings) of BCa. As it uses in most of the cases of classical white light, it is prone to error in tumor detection (rates of 10–20%) [31,32]. AI has the potential to alleviate human error when dealing with image interpretation.

Eminaga et al. [33] aimed to detect cancerous features from cystoscopy images using CNN models as instruments to perform diagnostic classification. In their case series, the Xception model performed the best (F1 score = 99.52%).

Lorencin et al. [34] used an ANN model using cystoscopy frames from 1997 images of BCa and 986 images of benign and normal-looking mucosa, and the results were very good, achieving an AUC of 0.99. The same team of researchers [35] utilized another CNN algorithm to separately identify benign from malignant lesions from 2983 images obtained during cystoscopy and achieved an AUC of 0.99. Ikeda et al. [36] developed a CNN algorithm obtaining good sensitivity and specificity (89.7% and 94%) for cancer detection in 2102 images obtained from cystoscopy. Yang et al. [37] used CNNs (LeNet, AlexNet, and GoogLeNet) and EasyDL platform to identify BCa images and found that EasyDL achieved the best accuracy (96.9%). Du et al. [38] used a CNN-DL algorithm to recognize BCa from 175 patients’ images, with the best accuracy with an EasyDL of 96.9%.

Shkolyar et al. [39] developed a CNN algorithm to identify tumors in an automated way (CystoNet). Images have been split into a training dataset (2335 images of normal bladder urothelium) and 417 images of papillary urothelial carcinoma from a total of 95 patients with a sensitivity of 91% and a specificity of 99%. Wu et al. [40] reported a diagnostic system based on cystoscopy images extracted with the ResNet 101 model and pyramid scene parsing network (PSPNet) framework resulting in higher accuracy and rapidness in reading and detection of BCa compared to expert urologists (accuracy = 93.9%, sensitivity = 95.4%). Ali et al. [41] aimed to study the sensitivity and specificity of photodynamic detection (PDD) blue-light cystoscopy and CNN algorithm to detect, grade, and stage BCa at cystoscopy, achieving a classification sensitivity and specificity for detection of BCa tumors of 95.77% and 87.84%, while the mean sensitivity and mean specificity of invasiveness of lesions are 88% and 96.56%, respectively. Yoo et al. [42] used AI SVM models and white light image (WLI) cystoscopy for detection and employed the red-green-blue method to grade BCa lesions. The sensitivity, specificity, diagnostic accuracy, and dice similarity coefficient (DSC) of AI were 95.0%, 93.7%, 94.1%, and 74.7%, respectively. For white light, the red and blue have been in accordance with tumor grade (*p* < 0.001), and the performance to distinguish between benign and low-grade lesions has been evidenced as 98% and >90% for detecting inflammatory lesions and CIS. Du et al. [38] used cystoscopy mages from 1002 normal bladder and 734 from bladder lesions and trained Caffe DL and EasyDL platforms to recognize BCa. Caffe DL obtained an accuracy of 82.9% and 96.9% on the EasyDL.

#### 3.2.2. Bladder Tumor Detection through Urine Cytology

Nojima et al. [43] used a 16-layer Visual Geometry Group (16VGG) CNN to predict if urinary cytology can identify malignant or high-grade lesions. The 16VGG CNN achieved excellent performance for differentiating cancer from benign tissue (AUC 0.9890 and an F1 score, 0.9002), and if the tumors were invasive BCa (AUC 0.8628 and an F1 score 0.8239) or high-grade BCa (AUC 0.8661 and an F1 score 0.8218). Awan et al. [44] developed a method to automatically identify atypical and neoplastic cells, and from all models, Xception performed best in their validation set (AUC 0.99). Vaickus et al. [45] aimed to identify the potential of a hybrid DL and morphometric model for the automation of the Paris System. Urine cytology whole slide images have been documented from 51 negative, 60 atypical, 52 suspicious, and 54 positive cases, achieving a 95% accuracy rate for the detection of cell type and atypia, and this can aid in automating the Paris System. Sanghvi et al. [46] reported results using a CNN algorithm that aimed to diagnose BCa on cytology images (2405 ThinPrep glass slides) and validated on a different data set and showed that the algorithm achieved an AUC of 0.88 (95% CI, 0.83–0.93), with a sensitivity of 79.5% and the specificity of 84.5% for high-grade urothelial carcinoma. Khosravi et al. [47] employed CNN methods to differentiate four biomarkers of BCa and four immunohistochemistry staining scores of BCa. The Inception V1-Fine tune algorithm achieved the best discrimination of the blood biomarkers with an accuracy of 99%. Sokolov et al. [48] scanned bladder cell surfaces using an imaging method (atomic force microscopy) involving multiple analyzed parameters to non-invasively detect BCa obtaining an accuracy of 94%, significantly higher than cystoscopy applied in 25 BCa and 43 control patients. Lilli et al. [49] used a DL model to detect cancer cells from urinary cytopathology images and found a weak performance of standard CNN algorithm used, and only after applying focal loss the model slightly improved accuracy and expected calibration error, up to 89.90% for urinary cytology.

#### 3.2.3. Bladder Tumor Detection through Urine Metabolomes

Some studies aimed to investigate the role of urine metabolomes in the detection of BCa as this comes into direct contact with the urine. Shao et al. [50] profiled 87 samples of BCa and 65 control samples and identified imidazoleacetic acid as a marker potentially related to BCa using an ML model and a decision tree (DT) obtaining accuracy of 76.60%, a sensitivity of 71.88%, and a specificity of 86.67%. Kouznetsova et al. [51] aimed to research the urine metabolites as biomarkers to classify BCa with the use of an ML model. Early stage BCa biomarker is D-glucose which is able to impact some potential neoplastic genes (AKT, EGFR, and MAPK3). Additionally, late-stage BCa-identified biomarkers (glycerol, choline, 13(S)-hydroxyoctadecadienoic acid, 2’-fucosyllactose, and insulin can have an important role as detection biomarkers. The best-performing model predicted metabolite class (accuracy of 82.54%, AUC of 0.84 on the training set).

#### 3.2.4. Bladder Cancer Segmentation Research

Bladder auto-segmentation, with the aim of differentiating the bladder wall from the surroundings, is mandatory to allow automatic diagnosis of bladder wall lesions. The bladder is a “shifting” organ in relation to its volume content, pressure, and physiology, by variations of urine content and lesion appearance. Therefore, some BCa can be excluded from the area of the region of interest (ROI), and other non-bladder-related images can be detected as bladder tumors. The current manual delineation executed by expert radiologists takes a long time and has great economic costs [52]. This makes segmentation challenging. The development of automated capabilities to segment the bladder in line with results obtained by the involvement of experts will alleviate the burdens in BCa research. Cha et al. in 2016 [53] aimed to segment regions of interest from the outside and inside and to detect bladder BCa from CT urography images from 173 patients (81 in the training set and 92 for validation) with a CNN algorithm. The algorithm performed better for segmentation performance compared to previously used methods. Exact segmentation of bladder walls and ROIs is mandatory for imaging detection of tumor stage and grade and has been studied by Dolz et al. using MRI and a CNN on 60 confirmed BCa patients. They obtained an accuracy of 0.98, 0.84, and 0.69 for the inner wall, outer wall, and tumor region segmentation, respectively [54]. Li et al. [55] proposed an automatic segmentation method on 1092 MRI images, showing that the DL U-Net method can show high accuracy results with a DSC of 85.48%. Niazi et al. [56] proposed a multi-class image segmentation method to discriminate between bladder layers in an automatic fashion (U-Net) for T1 histopathologically confirmed tumors and identified that a 12-layer model on hematoxylin-eosin stained images, achieved an accuracy of 89.3% ± 0.6 out of 100% for segmentation. Ma et al. [57] developed a U-Net DL model to segment the bladder through CT-urography using 81 patients for training and 92 patients for testing. The improvements for DCNN compared to U-Net DL have been statistically significant (*p* < 0.001). Zhang et al. [58] used AI to segment cystoscopic images. The analyzed attention mechanism-based cystoscopic images segmentation model indicated better performance in segmenting bladder tumors with a DSC of 82.7% and MIoU of 69% based on a U-Net DL-CNN model.

#### 3.2.5. Bladder Cancer Imaging and Artificial Intelligence

The most advancement in AI and urological cancer imaging is to be seen in prostate and renal neoplasia. AI and BCa imaging is yet limited, and the role of cystoscopy is still predominant in initial diagnosis. Several studies combined radiomics and AI to be able to recognize non-muscle invasive diseases due to the understaging of biopsies of about 50% [59]. Radiomics in prostate and renal cancers holds future promises, as seen in the latest research evidence, and relies on the analysis of images with the help of CAD and specific mathematical algorithms to obtain quantitative features not available to reader observers [60,61,62,63]. Imaging segmentation challenges have been addressed in Section 3.2.3. Xu et al. [64] aimed to analyze 3D texture features between bladder lesions and wall tissues. Using T2w MRI images from 62 cancer lesions and 62 volumes of interest (VOI), obtaining 29 features from recursive feature elimination-based support vector machine classifier (RFE-SVM) achieved good sensitivity, specificity, accuracy, and AUC and by augmentation synthetic minority oversampling technique improved the sensitivity, specificity, accuracy a AUCs values of 89.67%, 87.80%, 88.74% and 0.9416, respectively. Wu et al. [65] studied radiomic MRI features to predict lymph node metastasis in BCa with images obtained from 103 BCa individuals, with training and validation sets (69, 34, respectively). Features were obtained from T2w images, and the signature was employed using the least absolute shrinkage and selection operator (LASSO) algorithm and resulting in the ability to predict lymph node metastasis with good AUCs and the build nomogram (radiomics signature and lymph node status) pointed a good performance of calibration and discrimination in the training and validation sets (AUC of 0.9118 and of 0.8902, respectively). Zheng et al. [66] reported a radiomics and clinical nomogram aimed at pre-operative discrimination of muscle invasiveness of BCa 2602 radiomics features extracted using T2w MRI images. LASSO algorithm was used to build a radiomics signature for the training set, and a combined radiomics and clinical nomogram was developed and yielded good results (AUC of 0.913 for training and 0.874 for validation) and also demonstrated clinical usefulness. Kozikowski et al. [67] performed a systematic review and meta-analysis to identify the role of radiomics in the staging of BCa and predict its invasiveness in the muscle wall. The AUC of HSROC has been identified as 0.88, and the specificity and sensitivity in predicting invasiveness were 81% and 82%, respectively. Taguchi et al. [68] aimed to validate in a prospective multicenter study the vesical imaging and reporting data system (VI-RADS) using the latest MRI technology and DL reconstruction. This has been studied in 68 BCa patients and found that the accuracy of diagnosing muscle invasion using a cutoff of VI-RADS ≥ 4 was 94% (AUC 0.92) and DL reconstruction identified four further patients, initially misdiagnosed by VI-RADS score 3, and proper diagnosed was set by T2w imaging + denoising DL reconstruction. Sarkar et al. [69] used a hybrid ML and DL model to automatically detect and stage BCa and discovered that their LDA classifier on the XceptionNet platform had the best performance (accuracy = 86.07%, sensitivity = 96.75%, specificity = 69.65%, precision = 83.07% and F1-score = 89.39%) for detecting normal lesions from BCa. For detecting invasiveness, the same hybrid approach achieved medium results (accuracy = 79.72%, sensitivity = 66.62%, specificity = 87.39%, precision = 75.58%, and F1-score = 70.81%).

#### 3.2.6. Bladder Cancer Grading and Artificial Intelligence

Zhang et al. [70] aimed to describe texture features from MRI images to discriminate between low grade in 32 patients and high grade in 29 patients with BCa. The SVM classifier had the best performance in BCa grading (AUC, accuracy, sensitivity, and specificity of 0.861, 82.9%, 78.4%, and 87.1%, respectively). Wang et al. [71] developed and validated with the use of radiomics MRI a possibility of pre-operative grading of BCa in 31 high-grade and 39 low-grade patients. Radiomics features were extracted from T2w, DWI, and ADC images and analyzed by the LASSO algorithm. Multimodality models performed better (AUCs—max-out 0.9233, 95% CI 0.9001–0.9466; concatenation 0.9233, 95% CI 0.9001–0.9466) than single modality models (T2w, DWI and ADC) similar to the validation cohort. Jansen et al. [72] developed a fully automated grading DL system (U-Net segmentation network trained to detect bladder urothelium) that was able to grade 76% of the low-grade cancers and 71% of the high-grade cancers in accordance with the expert consensus.

#### 3.2.7. Bladder Cancer and Histopathology

Smith et al. [73], since 2011, aimed to construct a gene expression model for lymph node status prediction after cystectomy, utilizing a weighted nearest neighbor (WNN) classification algorithm rooted in Bayesian decision. They have identified in multivariate logistic regression that the model had a calculated AUC of 0.67 and was an independent predictor for lymph node involvement.

Seiler et al. [74] used a K-nearest neighbor classifier 51 (KNN51) classifier to predict pathological lymph node metastases in 199 cystectomy patients. Whole transcriptome expression profiles have been developed, and two cancer signature genes were used for comparison. The KNN51 classifier performed better than the comparison genes (AUC = 0.82, compared to 0.62 for 15-gene cancer recurrence signature (RF15) and 0.46 for 20-gene lymph node signature (LN20)) and has significant odds of predicting metastases in the lymph nodes compared to RF15 and LN20 (*p* < 0.001).

Wu et al. [75] reported the results of a nomogram combining genomic, clinical, and pathological data to predict lymph node status in 325 BCa patients using mRNAs from the TCGA database. They identified five mRNAs related to lymph node status and incorporated them into the nomogram, allowing a good discriminatory and lymph node status prediction ability with a logistic regression algorithm (AUC = 0.89).

Zhang et al. [76] proposed an automated way to diagnose on whole slide digital imagining of pathology results with the aid of AI. They automated analyzed 913 whole slide data images of BCa patients, and the CNN algorithm obtained similar results as the expert pathologists (AUC = 0.97). Velmahos et al. [77] extracted imaging biomarkers using histopathology slides to predict fibroblast growth factor receptor (FGFR) alterations in 418 BCa patients employing a CNN to identify tumor-infiltrating lymphocytes percentage for the prediction. The best model was achieved only on FGFR2/FGFR3 mutation, with a sensitivity of 82%, a specificity of 85%, and an AUC of 0.86.

#### 3.2.8. Bladder Cancer Staging and Artificial Intelligence

Garapati et al. [78] studied the feasibility of an automatic ML technique to screen 84 bladder cancer tumors assessed by CT urography, grouping lesions above or equal to T2 or below T2. Morphological and texture features alone or combined achieved comparable performance. Xu et al. [79] aimed to pre-operative stage BCa into non-muscle invasive or muscle invasive. Therefore, they used multiparametric MRI 1104 radiomics features from 54 patients to differentiate between the two entities. SVM-RFE and synthetic minority oversampling technique (SMOTE) have been employed to develop the model. A total of 19 features were analyzed from T2w and DWI sequences and performed better in muscle invasion discrimination AUC (0.9857). The model SVM-RFE+SMOTE classifier outperformed the experts in diagnostic accuracy (96.30%). Yin et al. [80] tried to differentiate Ta from T1 BCa on hematoxylin and eosin stain images from 1177 BCa tissues using an ML-CNN model imaging processing software Image J and Cell Profiler and had accuracies between 91% and 96%. Yang et al. [81] used a DL-CNN model to differentiate non-muscle from muscle-invasive BCa from 1200 CT images belonging to 369 patients. The best-performing model had an AUC of 0.997 and a sensitivity and specificity of 88.9% and 98.9%. Li et al. [82] aimed to assess the accuracy of radiomics, single-task DL, and multi-task DL on T2w MRI images to stage muscle invasiveness of BCa from 121 BCa lesions. AUCs were obtained for radiomics, single-task, and multi-task DL algorithms (0.920, 0.933, and 0.932, respectively). Li et al. [83] compared a DL-CNN model based on T2w with VI-RADS to predict muscle invasiveness of BCa and found that were higher AUCs for the DL model compared to two expert radiologists (AUC = 0.963, 0.843, and 0.852, respectively) and the accuracy was higher as the experts for VI-RADS 2 or 3 scores (*p* = 0.006). Xu et al. [84] developed a DL algorithm to detect and stage BCa on CT images of 60 patients having the disease. These images have been processed by the DL DCNN based on the You Only Look Once (YOLO) algorithm, and in the clinical staging, the coincidence rates with pathological results were found to be excellent (T1 stage = 50.01%, T2a = 91.65%, T2b, T3 and T4 stage = 100.00%) and not different from the clinical staging of pathological diagnosis (*p* > 0.05). Zou et al. [85] aimed to differentiate muscle from non-muscle invasive BCa and used T2w images with the Inception V3 platform to extract features and build a model (Multi-task BCa Muscular Invasion Prediction) and achieved the best results in the prospective data group (accuracy = 92.3%, sensitivity = 100%, and specificity = 88.5%). The summary of the studies has been embedded in Table 2.

## 4. Discussion

AI technology through ML and DL methods could have the potential to achieve improved outcomes for BCa patients. Several AI approaches in the different steps of BCa diagnostic work-up have been recently evaluated with the aim to improve oncologic and QoL outcomes of BCa patients, as well as to lower the financial burden related to BCa, which is the most expensive neoplasm to treat over the patients’ lifetime [86].

Our search retrieved 48 studies that are from all clinical tasks with an impact on BCa diagnosis, focusing, in particular, on the most recent evidence available in the current literature, therefore placing our work among the most comprehensive review dealing with AI and BCa. Specifically, in previous sections, we reviewed studies on ML- and DL-based tools applied to different diagnostics in BCa patients with regard to cystoscopy, urine cytology, urine metabolomes, bladder segmentation, imaging, grading, histopathology, and staging.

The retrieved studies were mostly focused on the use of AI methods applied to cystoscopy images for BCa detection [33,34,35,36,37,38,39,40,41,42]. Indeed, endoscopy plays a pivotal role in the initial diagnosis of BCa, since it allows direct visualization of bladder tumors and/or suspicious areas which deserve histopathological characterization. Although the combination of AI with cystoscopy is a relatively novel concept, AI methods bear the potential to mitigate human errors related to image interpretation (which is up to 20% for BCa detection). Due to their inherent design and nature, AI methods are immune to many errors clinicians have to face (e.g., fatigue, stress, burn-out, etc.) so that they can act as physician assistant tools for improved diagnostic performance during cystoscopy. To this purpose, several ML and DL models have been assessed, and in almost all cases, they outperformed conventional diagnostics, achieving the best AUCs. However, the studies retrieved to date were retrospective in design and mainly focused on methods applied to static frames taken during cystoscopy, which limits their clinical applicability since endoscopic diagnosis is a dynamic process (i.e., not depending only on images alone). Real-time AI-based tools and/or systems applied to endoscopic videos could solve this issue. Although full automation of endoscopic procedures is unlikely to occur anytime soon, advances in endoscopy for other malignancies (e.g., colonoscopy and esophagus-gastro-duodenoscopy) have achieved accurate automatic real-time detection of tumors, proving that the implementation of AI to traditional devices could add diagnostic value also for BCa detection. However, to the best of our knowledge, no study has prospectively reported on the real-time performance of AI approaches for BCa diagnosis during cystoscopy so far.

Although cystoscopy is the key to BCa diagnosis, it is an invasive procedure with significant implications in terms of healthcare costs for both patients and society and non-neglectable mental and physical discomfort for the patients. As a consequence, in recent years, there have been increasing efforts to train AI-based tools for urine sample analysis.

Urine cytology is a non-invasive test, yet on the other hand, it exhibits low sensitivity (up to 54%) for BCa diagnosis. Specifically, urine cytology has higher sensitivity in high-grade (79%) but low in low-grade tumors (79% and 16%, respectively) [42,87,88]. Thus, a negative cytology cannot rule out the presence of BCa. Moreover, urine cytology is user-dependent, and the evaluation can be hampered by low cellular yield, as well as urinary tract infections and/or stones; however, in experienced hands, specificity exceeds 90%. ML and DL methods have been used to automatically identify atypical and BCa cells and to improve the diagnostic accuracy of urine cytology, especially for low-grade cancers [43,44,45,46,47,48,49]. In this contest, two studies have also evaluated the use of AI-based tools for the assessment of urine metabolomes in identifying biomarkers potentially related to BCa [50,51]. However, it should be noted that AI algorithms applied to cystoscopy data generally showed higher accuracy for BCa diagnosis than those applied to urine samples.

Due to the previously mentioned pivotal role of endoscopy for the detection of BCa, the investigation of AI methods applied to imaging modalities such as CT and MRI scans for BCa initial diagnosis is still limited compared to other urological malignancies (e.g., prostate and kidney cancers) [64,65,66,67,68,69]. Although several studies combined radiomics and AI-based tools to improve detection, bladder segmentation, staging, and grading for BCa, data to date do not allow for generalizability.

Bladder segmentation (i.e., differentiating the bladder wall from surrounding tissues) is a crucial step in the BCa diagnostic work-up in the AI and CAD era. The bladder is a “shifting” organ in relation to its volume content, pressure, and physiology. Moreover, boundaries between the bladder wall and the surrounding soft tissues exhibit low contrast in images. Proper bladder segmentation encloses avoiding that some BCa can be excluded from the area of ROI (i.e., false negative cases), while non-bladder related images can be detected as BCa (i.e., false positive cases). Currently, manual delineation executed by radiologists is challenging, as it still takes a long time and places a great financial burden [52]. To overcome these issues, designing automated capabilities to segment the bladder in line with results obtained by the involvement of experts is pivotal to improving BCa management. Several studies evaluated the performance of different DL methods applied to CT-urography, MRI, hematoxylin-eosin stained, and cystoscopy images for proper bladder segmentation, and found positive results [53,54,55,56,57], yet still far for application into clinical practice.

Grading is a main aspect of BCa management for clinical decision-making since up to 30% of non-muscle invasive cases consist of high-grade BCa that can progress to muscle invasion as well as develop metastases [89,90,91,92,93,94]. Only a few studies evaluated the potential of applying AI technologies to MRI for BCa grading purposes [70,71]. Although the radiomics models outperformed the conventional modalities, the sample size was rather small, and the results need to be validated further. Moreover, to overcome potential grading errors occurring during the assessment of grade on transurethral resection of bladder tumor (TURBt) slides by pathologists—mainly due to inter- and intra-observer variability—a fully automated grading DL system was designed, which correctly graded more than 70% of low- and high-grade BCa cases [72].

AI has also been used for histopathological analysis of BCa with the aim of automating the process and achieving better reproducibility and efficiency. Specifically, an automated method to diagnose on whole slide digital imagining of pathology results with the aid of an AI algorithm obtained similar results as the expert pathologists [76]. Moreover, a few studies reported on the use of AI applied to pathology for the assessment of lymph node status after cystectomy, showing good results [73,74,75].

Proper staging is essential for therapeutic decision-making in patients with BCa, yet current tools are still hindered by the sub-optimal ability to correctly stage BCa. Histopathological analysis of specimens obtained by TURBt is the cornerstone of BCa diagnosis and staging; however, TURBt still retains inaccuracy in determining the status of muscle layer infiltration (up to 50% with T1 disease on TURBt have a muscle-invasive BCa). Image modalities, such as CT and MRI, can provide further staging information, yet they show sub-optimal performance in evaluating microscopic invasion (T1 versus T2 disease). Thus, to date, their main use is to assess locally advanced disease (≥T3b disease). Some studies addressed BCa staging and AI methods applied to CT and MRI imaging [78,79,80,81,82,83,84,85]. However, even if the AUCs were higher for the proposed new models than traditional approaches in almost all studies, there was a large amount of variability, and results need to be validated further to enhance their robustness and reproducibility.

Although AI has emerged as a powerful tool in the field of BCa management, potentially revolutionizing the current and future panorama in terms of early and accurate diagnosis as well as personalized treatment, there are still several shortcomings to consider.

One of the main limitations of AI in BCa is the availability and quality of data, algorithm bias, and interpretability of AI-generated results. AI algorithms require large and well-annotated datasets for training and validation; skewed or poorly representative datasets will lead to biases that could also be difficult to detect owing to the complex computations of AI tools. Efforts should be made to collect and curate high-quality datasets that represent the heterogeneity of bladder cancer, ensuring the robustness and generalizability of AI models. In the case of BCa, comprehensive datasets with good image quality are available on digital pathology and radiological imaging; conversely, cystoscopy images are scarce. Some preliminary studies utilizing DL algorithms, like CNNs, revealed potentially good results. The limitation to clinical integration is due to the dynamic of cystoscopy in obtaining images, and this cannot be easily standardized (like, e.g., radiology or histopathology) [34,36]. On the other hand, AI and BCa imaging is yet limited, and the role of cystoscopy is still predominant in initial diagnosis. As presented in previous specific paragraphs, promising results were achieved by AI applied to CT and MRI scans for BCa staging, grading, and bladder segmentation. However, to date, research in this field has not reached sufficient maturity for a tangible clinical benefit.

With specific regards to research methodology, performance analyses in the retrieved studies have been carried out mainly using sensitivity, specificity, accuracy, and AUC as performance metrics to assess the effectiveness of AI-based tools compared to traditional methods for BCa management (Table 2). However, to date, there is still substantial heterogeneity in study design, as well as algorithms design and definition of evaluated outcomes that makes meaningful comparison of quantitative analyses difficult to assess. Moreover, most AI technologies were trained, validated, and tested on the same dataset of cases; this has frequently led to a statistical phenomenon—namely “overfitting”- for which an algorithm works at its best on its own dataset, yet worst on novel ones.

Another intrinsic limitation is the interpretability and explainability of AI algorithms. Often, AI models function as black boxes, making it challenging to understand the reasoning behind their predictions. This lack of interpretability hinders trust and acceptance by clinicians who require transparent methods.

Ethical considerations are also of pivotal importance when implementing AI in BCa diagnosis and treatment. Indeed, analysis of confidential patient electronic health records raises legal and governance issues related to an individual’s right to privacy versus the potential benefit of research. Patient privacy, the need for informed consent, data security, algorithm bias, and the responsibility of healthcare professionals in validating and interpreting AI-generated results are among the main ethical concerns that should be considered and faced before the widespread implementation of AI in the healthcare system. Moreover, efforts should be made to identify and mitigate potential biases in AI algorithms that could disproportionately affect certain populations, ensuring equitable and unbiased BCa diagnosis and treatment.

Adoption challenges should be considered prior to the incorporation of AI-based diagnostic tools into the clinical workflow. This aspect could be a substantial barrier—especially in developing countries—due to logistical considerations mainly related to the massive computational power that is required to run complex ML algorithms and devices, balanced against the cost that the health care system can afford. Furthermore, practical issues over the integration with existing systems, as well as training requirements for healthcare professionals, are other obstacles that should be faced for proper real-life implementation.

Finally, scalability is crucial to achieving the clinical applicability of AI algorithms, data, and models in the BCa field. The ability of AI-based diagnostic tools to operate at the size, speed, and complexity required for BCa assessment is still suboptimal, so it represents a major obstacle to widespread clinical implementation. To date, difficulties in implementing these technologies in large-scale healthcare systems still exist, and robust infrastructures to support their widespread adoption are needed.

Despite these limitations, advances in technology and research can significantly improve the diagnosis and treatment of BCa.

First, AI will enhance diagnostic accuracy for the detection and classification of bladder cancer from various imaging modalities. AI has a transversal nature that makes these new intelligent technologies applicable to multiple disciplines (e.g., urology, radiology, pathology, and medical oncology). Indeed, in the era of increasing emphasis on precision medicine and patient-tailored management of diseases, AI bears an appealing potential for optimized risk stratification and personalized treatment planning in BCa patients. Specifically, the capability to integrate patient factors with clinical and multi-omics data, such as genomics, proteomics, and transcriptomics, to identify molecular signatures and biomarkers will help to predict responses to specific therapies allowing for personalized treatment selection and improving patient outcomes. Additionally, AI may be used as clinical decision support to analyze patient-specific data and provide tailored recommendations for treatment strategies, surveillance schedules, and follow-up plans. Finally, AI technologies, combined with telemedicine platforms, can enable remote monitoring of bladder cancer patients, real-time evaluation of symptoms, treatment response, and disease progression, and ultimately reduce healthcare costs, allowing timely interventions when required. In addition to the previously reported characteristics, generative models of deep learning are adaptable and could be used together in a hybrid approach, i.e., utilizing multiple deep basic learning models and integrations using generative and discriminative learning, permitting to potentially outperform the limitations of a single generative model [95]. Furthermore, the possibility to utilize and integrate AI algorithms with traditional imaging analysis has been proposed and starting to be used in different medical fields [96,97,98]. The evidence at this time, highlighted in this paper, points out that AI-based tools represent a catching and emerging research field in BCa diagnosis, which could lead to advanced diagnosis when applied to different steps (i.e., cystoscopy, urine samples analysis, bladder segmentation, imaging, and histopathology). However, when transposed into clinical settings, there are several limitations to their use and application (e.g., variability in study design, algorithms utilized, and training methods), which call for further research, including prospective, large data sets, as well as external validation of results. Additionally, the use of AI and ML models in common clinical practice is still limited by regulatory considerations as the pending FDA and CE approval as not all AI/ML-based medical devices have been approved, and the numerous and growing use of this technology further delays this process [99,100]. A further issue is linked to ML interpretability, which suffers from the lack of mature definitions and formality of methods and models, creating a sort of black box approach to AI in which the way of operation and process production are not known. This ambiguity and lack of transparency in the results obtained have therefore limited the adoption and extensive use of ML systems in sensitive domains [101]. Lastly, another potential limitation is related to the approach and the perspectives of patients toward AI, with controversial results in terms of support, understanding, and use of this technology in common clinical practice [102,103].

## 5. Conclusions

Artificial intelligence, together with its subsets, i.e., machine learning, deep learning, and artificial neural networks, has emerged as a powerful tool in the field of bladder cancer research and treatment, revolutionizing the current and future panorama in terms of early detection, accurate diagnosis, and personalized treatment. The large amount of data that could be analyzed via this technology could further improve the outcomes and the understanding of the disease, aiding patients and clinicians in common clinical practice. Additionally, the potentialities related to the use of machine learning and deep learning algorithms in developing novel biomarkers, drugs, and treatment protocols represent another pivotal point in the next future. Nevertheless, several challenges are still to be overcome, such as data quality, general applicability, and lastly, ethical considerations. Future studies are required in order to provide more consistent data regarding the role of this promising technology in the diagnostic and therapeutic pathway of bladder cancer.

## Figures and Tables

**Figure 1 diagnostics-13-02308-f001:**
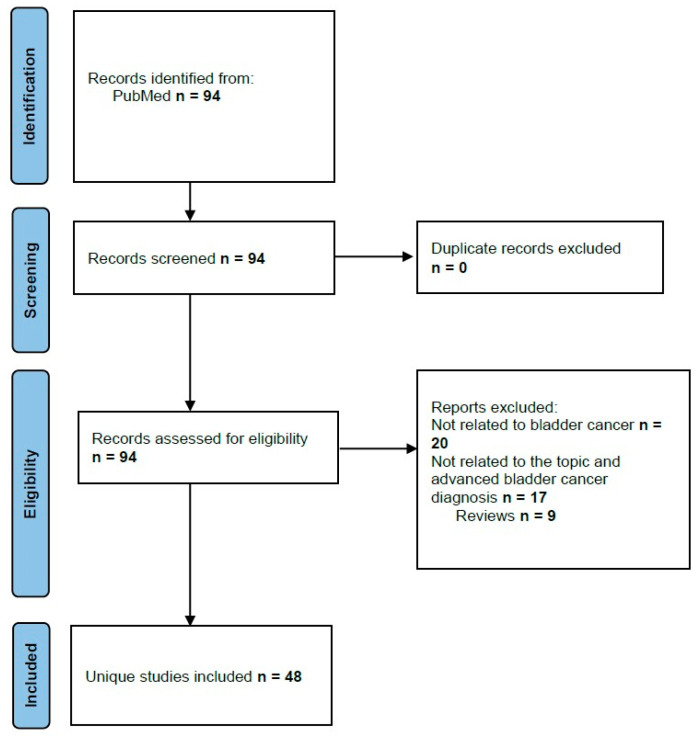
Flow chart of artificial intelligence and diagnosis of bladder cancer retrieval studies.

**Figure 2 diagnostics-13-02308-f002:**
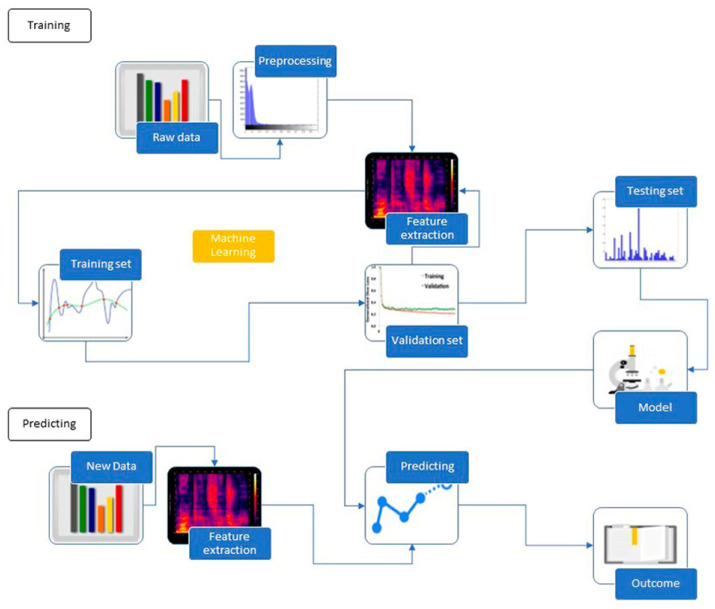
Summarization of machine learning process.

**Figure 3 diagnostics-13-02308-f003:**
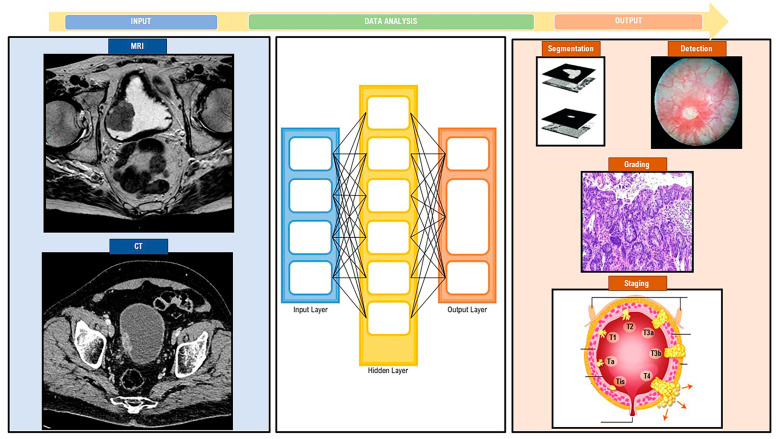
Deep learning workflow in bladder cancer.

**Table 1 diagnostics-13-02308-t001:** A glossary of deep learning workflow terms.

Term	Brief Explanation
Perceptron	a super-simplified version of a biological neuron, which takes different inputs and weighs them up to produce a single output [15]
Backpropagation	an algorithm that is used to train neural networks [15]
Artificial Neural Networks (ANN)	a computational model (i.e., algorithms or physical hardware) which mimics the human brain to process data and create patterns for decision-making [17]
Convolutional Neural Networks (CNN)	a neural network utilizing numerous identical copies of the same neuron, thus allowing a network to learn a neuron once and use it in several places. It is particularly useful for digitized images and pattern recognition [29]
Recurrent Neural Network (RNN)	a neural network utilizing sequential information, thus relying on previous computations [29]
Supervised Neural Network	a neural network for which, to produce an ideal output, a prior provided output is required. It is ‘trained’ on a given pre-defined dataset and provides outputs depending on the input it has received [30]
Unsupervised Neural Network	a neural network for which no labels are required. This involves giving a program with an unlabeled data set (i.e., that it has not been previously trained for). It is used to discover patterns and trends by clustering. [30]

ANN = Artificial Neural Networks; CNN = Convolutional Neural Networks; RNN = Recurrent Neural Networks.

**Table 2 diagnostics-13-02308-t002:** Summary of artificial intelligence used in studies for the advanced diagnosis of bladder cancer.

Authors/Year	INPUT/ N of Patients	AI Algorithm/ Models	OUTPUT	Summary	Performance
Smith et al., 2001 [73]	Gene expression Training: 156 pt Validation: 185 pt	WNN	Histopathology: pN stage	Using WNN to develop a gene expression model to predict pathological node status	AUC = 0.67
Seiler et al., 2016 [74]	Gene expression Training: 133 pt Validation: 66 pt	k-NN	Histopathology: pN stage	Using k-NN to develop a gene classifier to predict pathological lymph node metastasis in MIBC	AUC = 0.82
Cha et al., 2016 [53]	CT Urography Training: 81 pt Validation: 92 pt	CNN	Segmentation	Using CNN to segment bladder and ROIs	JSC: 0.76
Xu et al., 2017 [64]	T2w MRI images 62 cancer lesions, 62 controls	SVM	Histopathology: Presence of Cancer	Extracting Radiomics feature to differentiate cancer and non-cancer areas	AUC: 0.94
Shao et al., 2017 [50]	Urine metabolomes 87 BCa pt, 65 control	DT	Histopathology: Presence of Cancer	Evaluate urine metabolite associated with BCa	AUC: 0.77
Zhang et al., 2017 [70]	MRI radiomics features 61 pt	SVM	Histopathology: Grading	Using SVM to discriminate low grade and high grade bladder Ca on MRI	AUC: 0.86
Garapati et al., 2017 [78]	CT images texture analysis 76 pt	LDA CNN SVM RF	Histopathology: Staging	Comparing 4 AI algorithms to discriminate bladder Ca < T2 and ≥T2	AUC: 0.89–0.97
Vaickus et al., 2018 [45]	Urine cytology 51 negative, 60 atypical, 52 suspicious, and 54 positive cases	CNN (AlexNet/ResNet)	Citology: Detection	A hybrid deep-learning and morphometric algorithm to automate the PARIS system	ACC: >95%
Eminaga et al., 2018 [33]	Cystoscopy images	CNN	Cistoscopy: Detection	Detect cancerous features from cystoscopy images using CNN models	ACC: 0.99
Khosravi et al., 2018 [47]	IHC digital slides	CNN	Histopathology: Detection	differentiate 4 biomarkers of BCa on IHC	ACC: 0.99
Sokolov et al., 2018 [48]	High resolutions images using atomic force microscopy. 25 cancer lesions, 43 control	ML	Histopathology: Detection	Non-invasive detection of BCa	ACC: 0.94
Wu et al., 2018 [65]	T2w MRI images Training: 69 pt Validation: 34 pt	LASSO, LR	Histopathology: pN stage	Building a nomogram with mpMRI radiomic features	AUC: 0.84
Wu et al., 2018 [75]	Gene expression Training: 178 pt Validation: 246 pt	LR	Histopathology: pN stage	Utilizing LR to develop a genomic clinicopathologic nomogram for predicting LN metastasis	AUC: 0.89
Dolz et al., 2018 [54]	MRI images Training: 60 pt	CNN	Segmentation	Inner, outer wall, and tumor region segmentation	DSC: 0.69
Shkolyar et al., 2019 [39]	Cystoscopy images Training: 95 pt	CNN	Cistoscopy: Detection	Using “Cystonet” a CNN to discriminate malignant from benign images	SENS: 91% SPEC: 99%
Zheng et al., 2019 [66]	T2w MRI images Training: 130 pt Validation: 69 pt	LASSO, LR	Histopathology: pT stage	Building a nomogram with mpMRI radiomic features	AUC: 0.88
Wang et al., 2019 [71]	T2w MRI images Training: 70 pt Validation: 30 pt	LR	Histopathology: Grading	Utilizing MRI radiomics features to discriminate low and high-grade BCa	AUC: 88.2
Zhang et al., 2019 [76]	Histopathology digital images Training: 620 Validation: 193	CNN	Histopathology: pT stage	Utilizing CNN to analyze bladder Ca WSI compared to expert histopathologists	AUC: 0.97
Sanghvi et al., 2019 [46]	Urine cytology Training: 2405 urine sample Prospective Validation	CNN	Cistoscopy: Detection	Artificial Intelligence Algorithm for Reporting Urine Cytopathology	AUC: 0.88
Kouznetsova et al., 2019 [51]	Urine metabolomes	ANN, LR	Histopathology: pT stage	Recognition of Early and Late Stages of Bladder Cancer Using Metabolites and Machine Learning	ACC: 0.82
Ma et al., 2019 [57]	CT Urography Training: 81 pt Validation: 92 pt	U-net DCNN	Segmentation	Deep Learning Bladder Segmentation in CT Urography	JSC: 0.85
Xu et al., 2019 [79]	T2w and DWI MRI images Training: 54 pt	SVM	Histopathology: pT stage	BCa staging with MRI Radiomics Analysis	AUC:0.97
Ikeda et al., 2020 [36]	2102 Cystoscopy images	CNN	Cistoscopy: Detection	Development of a Support System for Cystoscopic Diagnosis of BCa	AUC: 0.98
Lorencin et al., 2020 [34]	2983 Cystoscopy images	ANN	Cistoscopy: Detection	Development of a Support System for Cystoscopic Diagnosis of BCa	AUC: 0.99
Li et al., 2020 [55]	MRI 1092 pt	U-net	Segmentation	Deep Learning Bladder Segmentation in MRI images	DSC: 0.85
Niazi et al., 2020 [56]	Histopathology digital images of pT1 pt	U-net	Segmentation	Deep Learning for bladder layers identification on Pathology images	ACC: 0.90
Yin et al., 2020 [80]	Histopathology digital images of pTa and pT1 pt	SVM, LR, RF, ANN	Histopathology: pT stage	Histopathological staging of BCa using different ML Approaches	ACC: 0.96
Jansen et al., 2020 [72]	Histopathology digital images	U-net	Histopathology: Grading	Detection and grading of BCa	ACC: 0.76
Lorencin et al., 2021 [35]	2983 Cystoscopy images	CNN	Cystoscopy: Detection	Development of a Support System for Cystoscopic Diagnosis of BCa	AUC:0.99
Nojima et al., 2021 [43]	Urine cytology	16-layer Visual Geometry Group CNN	Detection and Grading	DL diagnosis and grading of BCa using urine Cytology	AUC: 0.98, F1 score: 0.90 (Presence/Absence) AUC: 0.86, F1 score: 0.82 (Invasive/non invasive) AUC: 0.86, F1 score: 0.82 (low-grade/high-grade
Yang et al., 2021 [37]	Cystoscopy images	CNN	Cystoscopy: Detection	Comparisons of a Support Systems for Cystoscopic Diagnosis of BCa	ACC: 0.97
Awan et al., 2021 [44]	Urine cytology	CNN	Detection	Identification of atypic cells	AUC: 0.99
Yang et al. (2021b) [81]	CT Images, 1200 images from 369 pt	CNN	Histopathology: pT stage	DL to differentiate Muscle-Invasive BCa with CT	AUC: 0.99
Lilli et al., 2021 [49]	Urine cytology	CNN	Detection	Identification of Cancer cells	ACC: 89.90%
Du et al., 2021 [38]	Cystoscopy images 1736 pt	CNN EasyDL Caffe DL	Cystoscopy: Detection	Comparisons of a Support Systems for Cystoscopic Diagnosis of BCa	ACC = 82.9% (Caffe DL) ACC = 96.9% (EasyDL)
Taguchi et al., 2021 [68]	MRI images 68 pt	CNN	Detection	VI-RADS score and DL for BCa detection	AUC: 0.92
Velmahos et al., 2021 [77]	Histopathology digital images	CNN	Histopathology: FGFR alterations and tumor-infiltrating lymphocytes	Deep Learning to Identify Bladder Cancers with FGFR-Activating Mutations	AUC: 0.86
Ali et al., 2021 [41]	Blue light cystoscopy images	CNN	Cystoscopy: Detection Histopathology: Staging	Blue-light cystoscopy and CNN algorithm to detect, grade, and stage BCa	Detect-SENS = 95.77% SPEC = 87.84% Staging-SENS = 88% SPEC = 96.56%
Yoo et al., 2022 [42]	Cystoscopy images	SVM	Cystoscopy: Detection	Cystoscopic Diagnosis of BCa using a red-green-blue method.	SENS = 95.0% SPEC = 93.7% DSC = 74.7%
Wu et al., 2022 [40]	Cystoscopy images	CNN (ResNet)	Cystoscopy: Detection	Support Systems for Cystoscopic Diagnosis of BCa	ACC = 93.9%, SENS = 95.4%
Xu et al. [84] 2022	CT images 60 pt	CNN YOLO	Histopathology: pT stage	Predicting pT stage at pre-operative CT scan	CR: T1 stage = 50.01% T2a = 91.65%, T2b, T3 and T4 stage = 100.00%
Zou et al., 2022 [85]	T2w MRI images Prospective cohort	CNN Inception V3	Histopathology: pT stage	CNN to extract features and build a model predicting pT stage	ACC = 92.3% SENS = 100% SPEC = 88.5%
Zhang et al., 2023 [58]	Cystoscopy images	U-Net	Segmentation	Deep Learning Tumor Segmentation during cystoscopy	Dice = 82.7% MioU = 69%
Li et al., 2023 [82]	T2w MRI images	CNN LASSO SVM	Histopathology: pT stage	Accuracy of radiomics, single- and multi-task DL on T2w MRI images for staging	Radiomics-AUC = 0.920 Singletask = AUC = 0.933 Multitask = AUC = 0.932
Sarkar et al., 2023 [69]	CT	Hybrid ML and DL	Histopathology: Detection Staging	Hybrid ML and DL model to automatically detect and stage BCa	Detection: ACC = 86.07% Staging: ACC = 79.72%
Li et al., 2023 [83]	T2w MRI images	CNN VI-RADS	Staging	DL-CNN model based on T2w vs. VI-RADS in BCa staging	(CNN) AUC = 0.963 (VIRADS) AUC = 0.84

BCa = Bladder Cancer; pt = patients; WNN = weighted nearest neighbor; AUC =area under the curve; ACC = Accuracy; k-NN = K-nearest neighbor; LR = Logistic Regression; CNN = Convolutional Neural Network, ROI = Region of interest JSC = Jaccard’s coefficient of similarity; SVM = support vector machine classifier; DT = decision tree; LDA = linear discriminant analysis, NN = neural network, RAF = random forest classifier; IHC = immunohistochemistry; LASSO = least absolute shrinkage and selection operator; DSC = dice similarity coefficient; VI-RADS = vesical imaging and reporting data system; SENS = sensitivity; SPEC = specificity; YOLO = You Only Look Once algorithm; (mIoU) mean Intersection over Union.

## Data Availability

Not applicable.

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
