# Peer review of "Artificial Intelligence in the Advanced Diagnosis of Bladder Cancer-Comprehensive Literature Review and Future Advancement"

_diagnostics, 2023, doi:10.3390/diagnostics13132308_

Round 1

Reviewer 1 Report

1. Clarify the terminology: The review should provide clear definitions and explanations of AI subsets, such as machine learning, deep learning, and artificial neural networks, to ensure readers understand these concepts.

2. Discuss limitations: Consider addressing the challenges and limitations associated with applying AI in bladder cancer diagnosis, including data availability, algorithm bias, and interpretability of AI-generated results. Highlight imaging modalities and Provide a detailed overview of imaging modalities used in bladder cancer diagnosis and explain how AI techniques can enhance image analysis and interpretation.   3. Emphasize personalized treatment: Discuss the potential of AI in risk stratification and personalized treatment planning for bladder cancer patients, incorporating clinical data, genomics, and patient factors. 4. Address ethical concerns: Discuss ethical considerations related to AI in bladder cancer diagnosis, such as patient privacy, informed consent, and the responsibility of healthcare professionals in validating and interpreting AI-generated results.   5. Comprehensive performance analysis: Include a thorough analysis of performance metrics used in previous studies, such as sensitivity, specificity, and AUC, to quantitatively assess the effectiveness of AI compared to traditional methods.   6. Discuss adoption challenges: Address barriers and challenges in adopting AI-based diagnostic tools in clinical practice, including integration with existing systems, cost-effectiveness, and training requirements for healthcare professionals.

7. Differentiating from recent reviews: This comprehensive review distinguishes itself by conducting an extensive and up-to-date literature search, including recent studies, to provide the most comprehensive overview of AI advancements in bladder cancer diagnosis.

8. Figure 3. Deep learning workflow in bladder cancer. shall be improvised and in its present state is not suitable for the journal standards.

9. Figure 2. The summarization of the machine learning process can be revised and produced with the use of icons to improvise and make it attractive like a graphical representation. In the present form, it deems not fit and not necessary to explain the basics as it's not the objective of the review.

10. Address scalability: Discuss the scalability of AI-based diagnostic tools, including the potential challenges of implementing these technologies in large-scale healthcare systems and the need for a robust infrastructure to support their widespread adoption.

11. Explore hybrid approaches: Consider discussing the potential for combining AI technologies with traditional diagnostic methods, such as integrating AI algorithms with expert pathologists' assessments or combining imaging analysis with molecular biomarkers, to achieve even higher accuracy and reliability.   12. Highlight patient perspectives: Acknowledge the importance of patient perspectives and involvement in the development and implementation of AI-based diagnostic tools, considering their preferences, values, and potential concerns regarding the use of AI in their healthcare.   13. Address regulatory considerations: Discuss the regulatory landscape and considerations for AI-based diagnostic tools in bladder cancer, including FDA approval, CE marking, and the development of guidelines and standards to ensure safe and effective implementation   14. Discuss interpretability: Explore the importance of developing AI models that provide transparent and interpretable results, allowing clinicians to understand the underlying reasons behind the diagnostic predictions. Most reviews just mention the previous works and their outcomes. There is a need for authors to summarise and interpret the outcomes by collating similar studies carried out by researchers globally. 

The authors need to check for spell errors and grammatical errors in the manuscript. 

Author Response

  1. Clarify the terminology: The review should provide clear definitions and explanations of AI subsets, such as machine learning, deep learning, and artificial neural networks, to ensure readers understand these concepts.

We thank the reviewer for her/his comments. We updated the manuscript accordingly, specifying the definitions and brief basic explanation of the AI subsets concepts.

  1. Discuss limitations: Consider addressing the challenges and limitations associated with applying AI in bladder cancer diagnosis, including data availability, algorithm bias, and interpretability of AI-generated results. Highlight imaging modalities and Provide a detailed overview of imaging modalities used in bladder cancer diagnosis and explain how AI techniques can enhance image analysis and interpretation.  

We thank the reviewer for her/his comments and for helping to improve our paper. With specific regards to this comment, the manuscript was modified according to this suggestion

  1. Emphasize personalized treatment: Discuss the potential of AI in risk stratification and personalized treatment planning for bladder cancer patients, incorporating clinical data, genomics, and patient factors.

We thank the reviewer for her/his comment. The manuscript was modified according to this suggestion

  1. Address ethical concerns: Discuss ethical considerations related to AI in bladder cancer diagnosis, such as patient privacy, informed consent, and the responsibility of healthcare professionals in validating and interpreting AI-generated results.  

We thank the reviewer for her/his suggestion. We updated the manuscript accordingly.

  1. Comprehensive performance analysis: Include a thorough analysis of performance metrics used in previous studies, such as sensitivity, specificity, and AUC, to quantitatively assess the effectiveness of AI compared to traditional methods.  

We thank the reviewer for her/his suggestion and for giving us the opportunity to improve our paper. This aspect is now present in the manuscript.

  1. Discuss adoption challenges: Address barriers and challenges in adopting AI-based diagnostic tools in clinical practice, including integration with existing systems, cost-effectiveness, and training requirements for healthcare professionals.

We thank the reviewer for her/his suggestion. We updated the manuscript accordingly.

  1. Differentiating from recent reviews: This comprehensive review distinguishes itself by conducting an extensive and up-to-date literature search, including recent studies, to provide the most comprehensive overview of AI advancements in bladder cancer diagnosis.

We thank the reviewer for her/his comments. We updated the manuscript accordingly, specify how the novelty and the number of studies involved in this review permit to place our work among the most comprehensive work regarding AI and BCa.

  1. Figure 3. Deep learning workflow in bladder cancer. shall be improvised and in its present state is not suitable for the journal standards.

We thank the reviewer for her/his suggestions. We improved the manuscript accordingly, modifying the figure which, however, is similar, in terms of content to other deep learning workflow reported in the literature. The idea was to maintain a schematic approach to the figures.

  1. Figure 2. The summarization of the machine learning process can be revised and produced with the use of icons to improvise and make it attractive like a graphical representation. In the present form, it deems not fit and not necessary to explain the basics as it's not the objective of the review.

We thank the reviewer for her/his suggestions. We improved the manuscript accordingly, modifying the figure which, however, is similar, in terms of content to other deep learning workflow reported in the literature. The idea was to maintain a schematic approach to the figures.

  1. Address scalability: Discuss the scalability of AI-based diagnostic tools, including the potential challenges of implementing these technologies in large-scale healthcare systems and the need for a robust infrastructure to support their widespread adoption.

We thank the reviewer for her/his suggestion. We updated the manuscript accordingly as suggested

  1. Explore hybrid approaches: Consider discussing the potential for combining AI technologies with traditional diagnostic methods, such as integrating AI algorithms with expert pathologists' assessments or combining imaging analysis with molecular biomarkers, to achieve even higher accuracy and reliability.

We thank the reviewer for her/his suggestions. We improved the manuscript accordingly.  

  1. Highlight patient perspectives: Acknowledge the importance of patient perspectives and involvement in the development and implementation of AI-based diagnostic tools, considering their preferences, values, and potential concerns regarding the use of AI in their healthcare.  

We thank the reviewer for her/his suggestions. We improved the manuscript accordingly.  

  1. Address regulatory considerations: Discuss the regulatory landscape and considerations for AI-based diagnostic tools in bladder cancer, including FDA approval, CE marking, and the development of guidelines and standards to ensure safe and effective implementation  

We thank the reviewer for her/his suggestions. We improved the manuscript accordingly.  

  1. Discuss interpretability: Explore the importance of developing AI models that provide transparent and interpretable results, allowing clinicians to understand the underlying reasons behind the diagnostic predictions. Most reviews just mention the previous works and their outcomes. There is a need for authors to summarise and interpret the outcomes by collating similar studies carried out by researchers globally.

We thank the reviewer for her/his suggestions. We improved the manuscript accordingly.  

Comments on the Quality of English Language

The authors need to check for spell errors and grammatical errors in the manuscript. 

We thank the reviewer for her/his suggestions. We improved the manuscript accordingly.  

Reviewer 2 Report

This paper is a comprehensive literature review on the use of artificial intelligence (AI) in the diagnosis of bladder cancer. The paper discusses how AI can improve prediction algorithms and enhance the performance of diagnosis, prognosis and treatment of urological diseases, including bladder cancer. The paper also focuses on the potential of AI in clinical practice and the challenges that still need improvements for a wide clinical integration in optimal cancer care.

The paper is well-written and organized, I suggest accept in the present form.

Minor editing of English language required

Author Response

We thank the reviewer for her/his comments and kind words. We updated the manuscript accordingly for English language corrections.